# Changes in Nutrient Concentrations in Shenzhen Bay Detected Using Landsat Imagery between 1988 and 2020

Jingjing Huang [1], Difeng Wang [1,2,*], Fang Gong [1,2,3], Yan Bai [1,2,3] and Xianqiang He [1,2]

1 State Key Laboratory of Satellite Ocean Environment Dynamics, Second Institute of Oceanography, Ministry of Natural Resources of the People's Republic of China, Hangzhou 310012, China; leowatson@sio.org.cn (J.H.); gongfang@sio.org.cn (F.G.); baiyan@sio.org.cn (Y.B.); hexianqiang@sio.org.cn (X.H.)
2 Southern Marine Science and Engineering Guangdong Laboratory (Guangzhou), Guangzhou 510000, China
3 National Earth System Science Data Center, Beijing 100101, China
* Correspondence: dfwang@sio.org.cn

**Abstract:** Shenzhen Bay (SZB), situated between Shenzhen and Hong Kong, is a typical bay system. The water quality of the bay is notably affected by domestic and industrial discharge. Rivers and various types of drainage outlets carry terrestrial pollutants into SZB, resulting in elevated concentrations of nitrogen and phosphorous as well as relatively poor water quality. For over 200 years, Hong Kong has practiced oyster farming within brackish estuarine waters. Oyster farming is a type of mariculture which includes oyster breeding in oyster rafts. Remote sensing is a monitoring technique characterized by large spatial coverage, high traceability, and low cost, making it advantageous over conventional point-based and ship-borne monitoring methods. In this study, remote-sensing models were established using machine-learning algorithms to retrieve key water-quality factors (dissolved inorganic nitrogen (DIN) and orthophosphate-phosphorous (PO4_P) concentrations, $C_{DIN}$ and $C_{PO4\_P}$, respectively) from long-term time-series data acquired by the Landsat satellites. (1) Spatially, the water quality in Inner SZB was worse than that in Outer SZB. (2) The water quality temporarily deteriorated between the end of the 20th century and the beginning of the 21st century; then it gradually improved in the late 2000s. (3) Monitoring the water quality in an oyster-farming area revealed that oyster farming did not adversely affect the water quality. (4) The result of monitoring the water quality in river estuaries in SZB shows that water quality was mainly affected by river input.

**Keywords:** Shenzhen Bay; remote-sensing monitoring; support-vector-machine model; backpropagation neural network

## 1. Introduction

Bays are important areas for marine-based economic activity, tourism, and mariculture. However, these activities lead to pollution in bays [1], such as in Shenzhen Bay (SZB), a typical bay in the northern South China Sea between Shenzhen and Hong Kong. Daily domestic activities and industrial operations (e.g., wastewater discharge from large industrial enterprises, maritime shipping, and aquaculture) put enormous pressure on coastal waters [2]. According to the 2012–2019 Shenzhen Municipal Environmental State Bulletins, the seawater quality in the western coastal waters near Shenzhen was lower than the Class IV standard of the Chinese Sea Water Quality Standard (GB 3097-1997) [3]. Dissolved inorganic nitrogen (DIN) and orthophosphate-phosphorus (PO4_P) were the primary pollutants [4]. The government of Shenzhen has launched a campaign to reduce pollution levels in SZB, joining forces with the government of Hong Kong to restore the Shenzhen River—the largest river that empties into SZB [5]. Water-quality monitoring can facilitate assessment of the current status of SZB as well as provide an evaluation of the cleanup measures.

Conventionally, water-quality factors (WQFs) are monitored through ship-borne and buoy-based measurements, sampling at offshore monitoring stations, or field surveys. These techniques can only acquire data at discrete spatial points and are relatively time-consuming and costly. In addition, it is difficult to use these techniques to conduct measurements over large spatial areas. In comparison, satellite remote-sensing technology is known for its advantages of high spatial coverage, temporal continuity, and low cost. One major application of remote sensing is to establish physical, empirical, and semi-empirical models between spectral information received by sensors and WQFs; retrieval methods are based on the principle that waters with different compositions exhibit different spectral characteristics [6]. Previous studies have shown that remote-sensing technology can be applied to the retrieval of nutrient concentrations in water. For example, some researchers use MODIS data to estimate the concentration of dissolved inorganic nitrogen ($C_{DIN}$) [7] and total phosphorus ($C_{TP}$) [8], in which the coefficient of determination $R^2$ is equal to 0.82 and 0.68, respectively. Du et al. used Sentinel-3 data with the same spatial resolution as MODIS to retrieve the $C_{TP}$ of Taihu Lake, achieving a root-mean-square error (RMSE) of 0.04 mg/L [9]. Some researchers also use higher spatial resolution satellite data, such as Landsat [10], WorldView-2 [11], and Sentinel-2 [12] to retrieve nutrient concentrations and apply them to the direct or indirect retrieval of total nitrogen ($C_{TN}$), $C_{TP}$, $C_{DIN}$, and dissolved inorganic phosphorus ($C_{DIP}$). Machine learning (ML) algorithms, including neural networks (NN) and support vector machines (SVM), have also been applied to the inversion of nutrient concentrations. Liu et al. established a retrieval model for $C_{TP}$ using an SVM model, which yielded a retrieval accuracy ($R^2$) of 0.604 [13]. Ding et al. estimated $C_{TP}$ with an artificial neural network and achieved a retrieval accuracy ($R^2$) greater than 0.73 [14]. Jiang et al. found that the extra-trees regression algorithm was the most suitable for the inversion of $C_{TN}$ in the Miyun Reservoir and yielded an absolute square error of 0.000065 [15]. Sinshaw et al. established a retrieval model for summertime $C_{TN}$ and $C_{TP}$ with pH, electrical conductivity, and turbidity as the input parameters for an NN and found that the pH had the highest sensitivity [16]. The above research shows the feasibility of sensing technology for nutrient-concentration retrieval, and it also reflects the advantages of ML methods for such retrieval in coastal waters [13–16].

Researchers have investigated water-quality inversion in Shenzhen Bay and adjacent waters. For example, Hafeez and Nazeer et al. retrieved the concentrations of suspended sediment concentration ($C_{SS}$) and chlorophyll-a ($C_{Chl-a}$) in the coastal waters offshore of Hong Kong from data acquired by the Landsat TM and ETM+ sensors as well as the HJ-1 A/B satellites [17,18]. Nazeer and Hafeez retrieved $C_{SS}$ and $C_{Chl-a}$ in the region extending from the Pearl River Estuary to the coastal waters of Hong Kong [19,20]. Liu et al. established a linear model for retrieving $C_{PO4\_P}$ in SZB and analyzed the changes based only on $C_{PO4\_P}$ retrieved from remote-sensing data for 2 years (1988 and 2009) instead of a continuous time series [21]. In summary, remote sensing research into nutrient concentrations in the region is mainly conducted through indirect inversion of $C_{SS}$ or $C_{Chl-a}$ data, and there is a lack of long-term time-series analysis. Therefore, we lengthened the time series, used remote-sensing technology to statistically analyze the changes in nutrients in SZB from 1988 to 2020 (more than 30 years), and explored the sources of nutrients.

In this study, several ML methods were used to establish retrieval models with optimum modeling signals based on the relationships between the remote-sensing data acquired by two satellites (Landsat-5 and Landsat-8) and in situ measurements of nutrient concentrations. By comparing the retrieval accuracies of the models established using various ML methods, optimum $C_{DIN}$ and $C_{PO4\_P}$ retrieval models were determined. These models were subsequently used to obtain the characteristics of the water-quality changes in SZB over the 32-year period from 1988 to 2020. In addition, focus was placed on determining the water quality in the oyster-farming area (OFA) and the estuary of the Shenzhen River in SZB. The goal was to examine the relationships between the overall water quality in SZB and the water quality in the OFA and the estuaries through which terrestrial matter is carried into SZB.



Therefore, using remote-sensing technology to retrieve the nutrient concentrations in SZB over the past 30 years, we obtain a more detailed understanding of the water quality in the area. We can also explore the sources of pollutants using the spatial pattern of nutrient concentrations. A back-propagation neural network (BPNN) [22], SVM, and high-resolution remote-sensing data in coastal second-class water bodies provide a basis for subsequent research in other bays.

## 2. Materials and Methods

### 2.1. Study Area

Situated between the Hong Kong Special Administration Region and Shenzhen, Guangdong Province (China), SZB (113°55′8″–114°5′13″E, 22°26′23″–22°30′46″N) is a semi-enclosed shallow bay (see Figure 1 for location). SZB is traversed by a bridge, and it is surrounded by urban land on three sides. The bay receives terrestrial input from rivers such as the Shenzhen River, the Dasha River, and the Shan Pui River. With a coastline of approximately 60 km in length, SZB encompasses an area of approximately 90.8 km². According to its geographic location, we define the northeast corner near the Shenzhen River and the Shan Pui River as Inner SZB; the southwest corner connected to the open sea is designated Outer SZB. For the analysis, we used a rectangular area of the same size (see Figure 2) to represent Inner SZB and Outer SZB. The water is relatively shallow in Inner SZB (depth generally less than 5 m and averaging 3 m), but it deepens in Outer SZB. In SZB, there are irregular semidiurnal tides with an average tidal range of 1.36 m at the mouth as well as southwest–northeast reversing tidal currents [23,24].

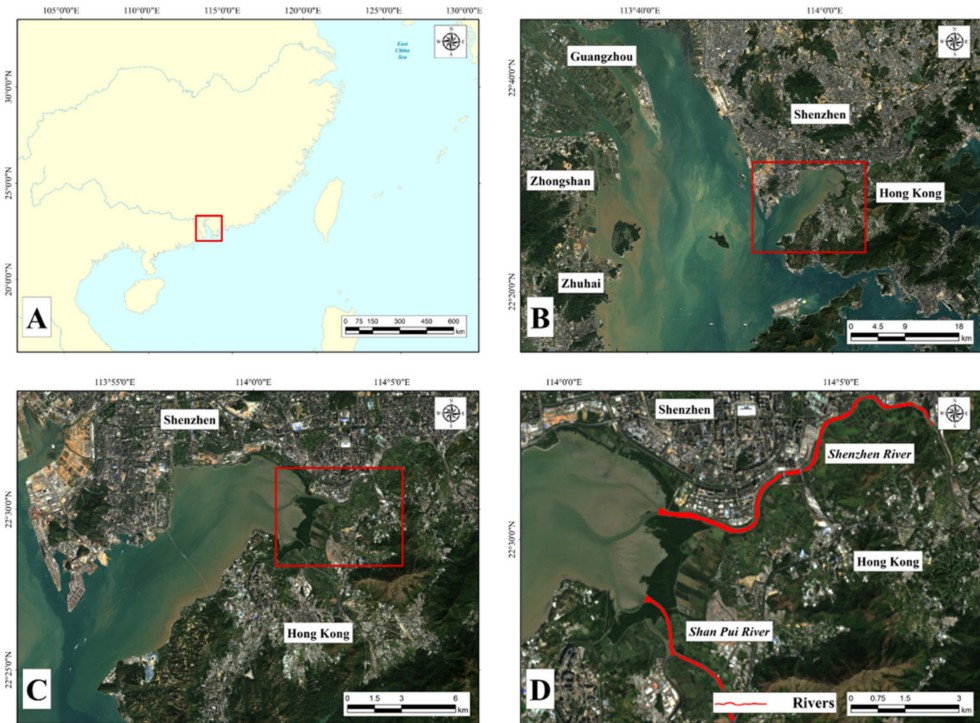

**Figure 1.** Schematic of the study area. (**A**) The approximate location of the study area (red box) in China. (**B**) Satellite image of the general location of the study area, where SZB is within the red box. (**C**) Close-up view of the study area: Shenzhen River and Shan Pui River are located within the red box. (**D**) A view of Shenzhen River and Shan Pui River.

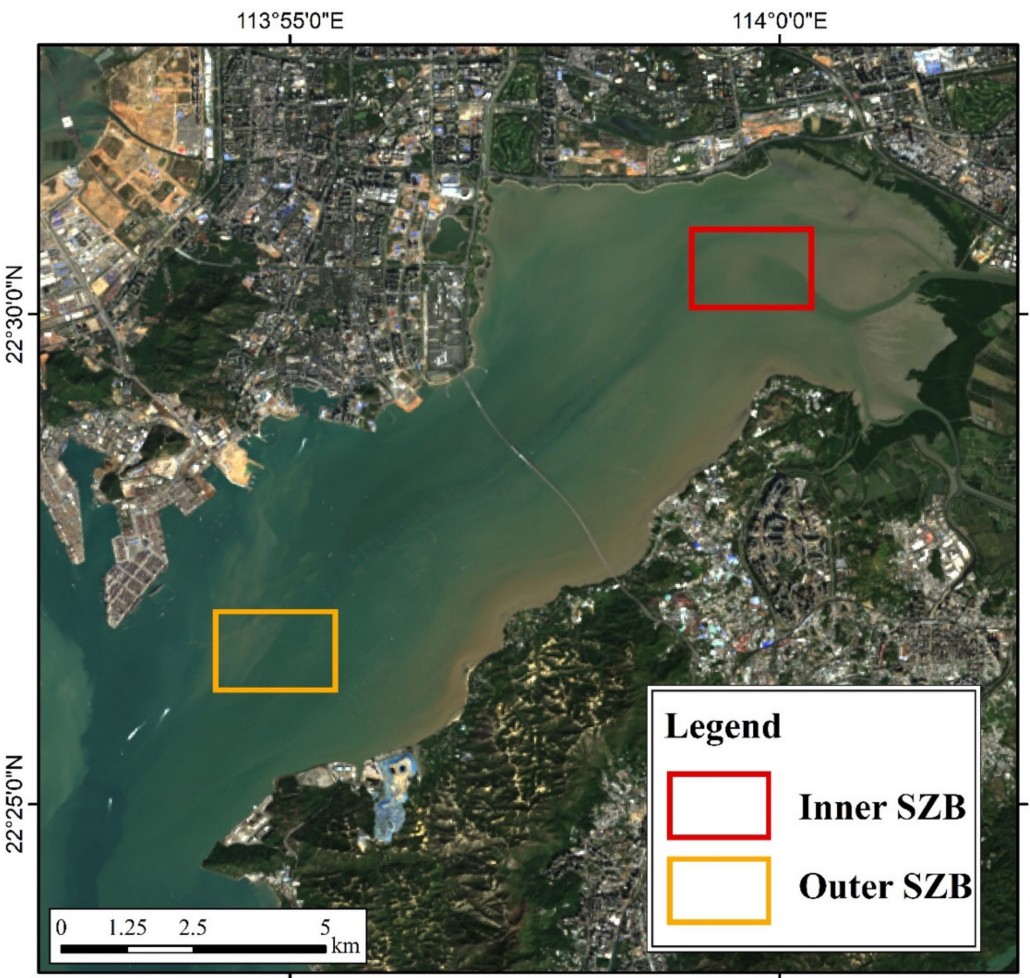

**Figure 2.** Representative analysis areas of Inner SZB and Outer SZB. The red box represents the Inner SZB water body, and the yellow box represents the water body of Outer SZB.

*2.2. In Situ Measurement Data*

The measurement data used in this study were obtained from the Environmental Protection Department of Hong Kong (HKEPD). The HKEPD divides the coastal waters of Hong Kong into ten control zones [25]: the Southern Zone, the Tolo Harbor and Channel Zone, the Northwestern Zone, the Victoria Harbor Zone, the Mirs Bay Zone, the Junk Bay Zone, the Eastern Buffer Zone, the Western Buffer Zone, the Port Shelter Zone, and the Deep Bay (i.e., SZB) Zone, where 76 water-quality monitoring points have been established. Monthly measurements have been continuously recorded at the monitoring points from 1986 to the present. The data are published on the website of the HKEPD (https://cd.epic. epd.gov.hk/EPICRIVER/marine/?lang=en, accessed on 1 December 2020). For this study, the measurement data at five sample stations in SZB were used (see Figure 3). Based on the statistics of the $C_{DIN}$ and $C_{PO4\_P}$ measured in situ in SZB from 1986 to 2019, $C_{DIN}$ ranges 0.11–14.05 mg/L, and the multi-year average is 2.15 mg/L. $C_{PO4\_P}$ is 0.001–2.9 mg/L, and the multi-year average is 0.222 mg/L.

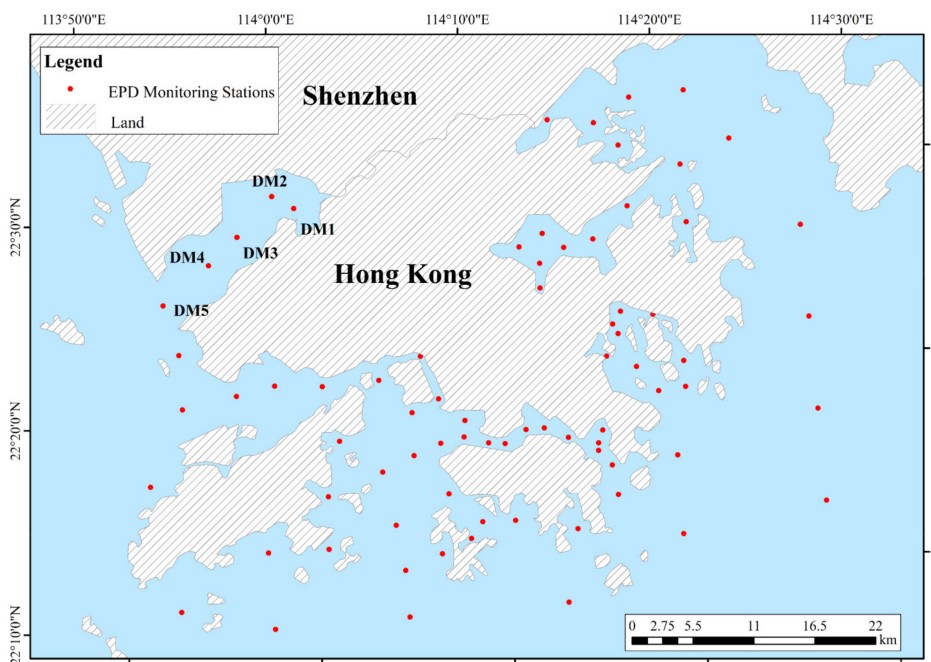

**Figure 3.** Locations of the 76 monitoring points in the ten water-quality control zones established by the HKEPD. Stations DM1 to DM5 were used in this study.

### 2.3. Remote-Sensing Data and Preprocessing

The remote-sensing data used in this study were acquired by the TM sensor onboard the Landsat-5 satellite and the OLI onboard the Landsat-8 satellite. The Landsat-5 and Landsat-8 satellites were launched by the United States National Aeronautics and Space Administration on 1 March 1984 and 11 February 2013, respectively. Landsat-5 TM has six bands with a spatial resolution of 30 m, including three visible bands, two near-infrared bands, and one mid-infrared band. Compared with Landsat-5, Landsat-8 has an additional coastal/aerosol band; the wavelength range of each band also differs. In this study, images with cloud coverage of less than 30% were selected. The path and row numbers for the study area were 122 and 44, respectively. In total, there were 37 Landsat-5 TM images (acquired between 1988 and 2011) and 22 Landsat-8 OLI images (acquired between 2013 and 2020) available for the study area (see Table 1). The data were downloaded from the United States Geological Survey Earth Explorer website (https://earthexplorer.usgs.gov/, accessed on 5 December 2020).

The images were subjected to preprocessing, including radiometric calibration and atmospheric correction (fast line-of-sight atmospheric analysis of spectral hypercubes (FLAASH) method). Previous studies have shown that the FLAASH method is effective for measuring the surface reflectance of water components [26].

**Table 1.** List of remote-sensing images.

| Path | 122 | Row | 44 |
|---|---|---|---|
| **Landsat-5** | | **Landsat-8** | |
| 24 November 1988 | 4 July 2000 | 9 August 2013 | 18 February 2020 |
| 10 December 1988 | 21 August 2000 | 29 November 2013 | 16 November 2020 |
| 11 November 1989 | 18 January 2003 | 31 December 2013 | 2 December 2020 |
| 13 December 1989 | 13 July 2003 | 16 January 2014 | |
| 5 December 1992 | 17 October 2003 | 15 October 2014 | |
| 22 September 1994 | 19 October 2004 | 16 November 2014 | |
| 8 October 1994 | 22 October 2005 | 3 January 2015 | |
| 25 November 1994 | 11 February 2006 | 19 January 2015 | |
| 9 September 1995 | 10 November 2006 | 18 October 2015 | |
| 3 March 1996 | 13 January 2007 | 7 February 2016 | |
| 7 June 1996 | 29 January 2007 | 26 March 2016 | |
| 9 July 1996 | 26 July 2008 | 5 November 2016 | |
| 25 May1997 | 15 November 2008 | 7 December 2016 | |
| 13 June 1998 | 17 October 2009 | 20 August 2017 | |
| 15 July 1998 | 4 December 2009 | 23 October 2017 | |
| 4 November 1998 | 13 March 2011 | 12 February 2018 | |
| 7 January 1999 | 1 June 2011 | 1 April 2018 | |
| 8 February 1999 | 4 August 2011 | 19 May 2018 | |
| 9 December 1999 | | 14 November 2019 | |

Water bodies were identified using the normalized difference water index (*NDWI*) [27].

$$NDWI = (R_{Green} - R_{NIR})/(R_{Green} + R_{NIR}) \tag{1}$$

where $R_{Green}$ represents the reflectance of the green band, and $R_{NIR}$ represents the reflectance of the *NIR* band. If the *NDWI* value is greater than 0, then the pixel is judged to be a body of water.

### 2.4. Method of Model Building

The optical properties of Case-2 waters are affected by colored dissolved organic matter (CDOM) and inorganic mineral particles, in addition to phytoplankton [28,29]. Due to the complex composition of Case-2 waters, models were established using ML methods, including SVM and BPNN.

BPNN is one of the most commonly used artificial neural networks, which is based upon the biological neural system in a highly simplified form. These networks provide a statistical tool for simulating the dependent variables of various engineering problems, especially where highly complex relationships define the physical processes of the problem [30]. The simplest three-layer structure of BPNN includes an input layer, hidden layer, and output layer (see Figure 4).

SVM theory is an approximation of the principle of structural risk minimization that involves the same processing complexity for high- and low-dimensional samples. Kernel functions are introduced to realize nonlinear mapping, thereby perfectly solving nonlinear processing problems and ensuring that SVMs are able to produce relatively good results when trained on a small number of samples [31]. The SVM model has been proven more suitable for modeling small sample data in previous research [13,31].

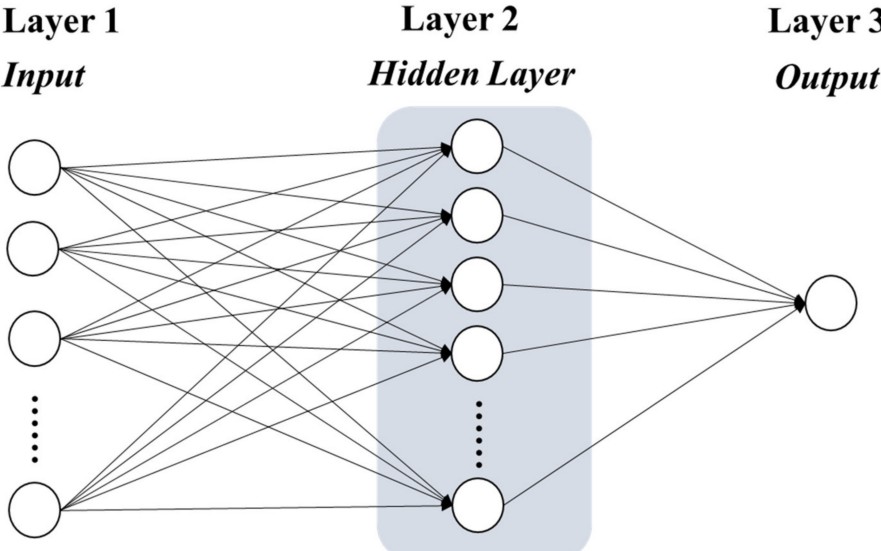

**Figure 4.** Three-layer BPNN structure.

In this paper, the optimum models for retrieving nutrient concentrations were determined by comparing model results from each ML method against empirical/in situ measurement data. Subsequently, the results were used to retrieve nutrient concentrations.

### 3. Results

*3.1. Modeling*

Corresponding radiance values in the remote-sensing images were matched based on the latitude and longitude of the HKEPD in situ data for SZB. To ensure synchronicity, the remote-sensing images acquired within 4 days of the measurement dates were selected. Given that the band settings for the Landsat-5 TM sensor and Landsat-8 OLI were not completely identical (the latter adds a new coastal band and the wavelength range of each band is slightly different), the synchronous datasets were separately matched. In total, 164 sets of valid radiance acquired by the Landsat-5 TM sensor from 1988 to 2011 were obtained. The corresponding in situ $C_{DIN}$ measurements ranged from 0.21 to 12.02 mg/L, averaging 2.30 mg/L. $C_{PO4\_P}$ ranged from 0.001 to 1.6 mg/L, averaging 0.22 mg/L. For the Landsat-8 OLI sensor, 25 sets of valid radiance values were obtained. The corresponding $C_{DIN}$ measurements ranged from 0.21 to 4.37 mg/L, averaging 1.52 mg/L. $C_{PO4\_P}$ ranged from 0.001 to 0.23 mg/L, averaging 0.1 mg/L.

Following data preparation, the inversion model was built. The correlations between radiance values from the remote sensing (RS) data and the in situ measurements are presented in Figure 5. The sensitive bands of LT5 and LT8 are different. For LT5, the most sensitive band for $C_{DIN}$ and $C_{PO4\_P}$ is the near-infrared band, followed by the red band. For LT8, the most sensitive band is the red band, followed by the green band. The in situ $C_{DIN}$ and $C_{PO4\_P}$ measurements are highly synchronous (Figure 6).

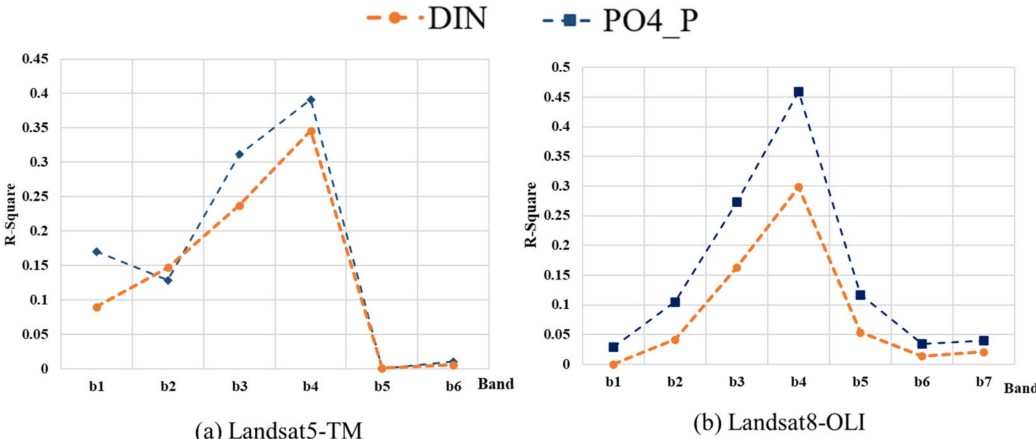

**Figure 5.** Correlations between the radiance values of Landsat single-band data and the in situ $C_{DIN}$ and $C_{PO4\_P}$: (**a**) Landsat-5 TM; (**b**) Landsat-8 OLI.

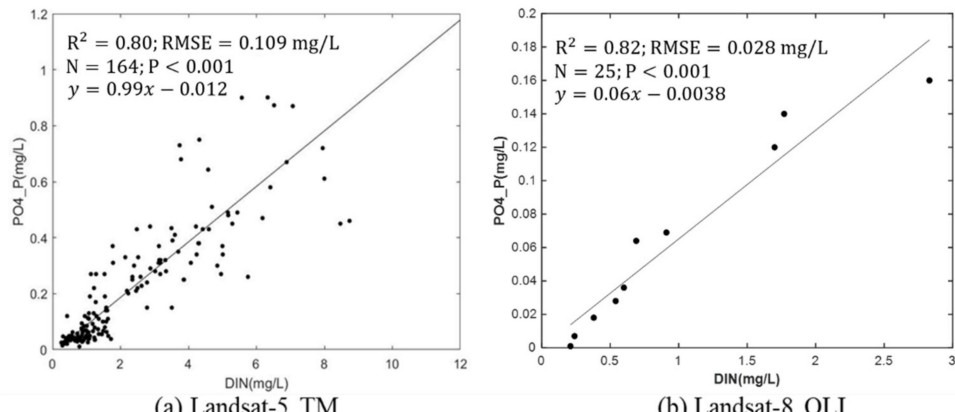

**Figure 6.** Analysis of the correlations between the in situ $C_{DIN}$ and $C_{PO4\_P}$ measurements (units: mg/L): (**a**) Landsat-5 TM; (**b**) Landsat-8 OLI.

Following identification of the relatively sensitive bands, multiple combinations of bands were tested. The nutrient concentrations derived from multiple combinations of bands were correlated with the band ratios and their measurements. Finally, the most suitable form of index factors was determined.

1.　Single-band form;
2.　Multi-band form;
3.　Band-ratio (combination) form.

Compared to the values derived from the remote-sensing data in single bands (see Figure 5), values derived from the remote-sensing data in combinations of bands were more strongly correlated with the in situ measurements (Table 2). Nutrient-concentration retrieval models for the two satellites were established based on the SVM and the BPNN models. The data were divided into a training set (80%) and a validation set (20%). Because the amount of data matched by OLI was small, the SVM method that was more suitable for small dataset modeling was given priority [31].

**Table 2.** R-square between the reflectance values based on Landsat-5 TM and Landsat-8 OLI data for various combinations of bands and in situ measurements of $C_{DIN}$ and $C_{PO4}$.

| Landsat-5 TM | DIN | PO4_P | Landsat-8 OLI | DIN | PO4_P |
|---|---|---|---|---|---|
| b(Blue), b(Green), b(Red), b(*NIR*) | 0.58 | 0.65 | b(Coastal), b(Blue), b(Green), b(Red), b(*NIR*) | 0.78 | 0.84 |
| b(Blue)/b(Red) | 0.39 | 0.42 | b(Red)/b(Coastal) | 0.56 | 0.57 |
| b(Blue)/b(*NIR*) | 0.30 | 0.26 | b(Coastal), b(Red) | 0.55 | 0.64 |
| b(Green)/b(Red) | 0.34 | 0.34 | b(Red), b(*NIR*) | 0.50 | 0.66 |
| b(Green)/b(*NIR*) | 0.22 | 0.18 | b(Coastal)/b(Red) | 0.47 | 0.51 |
| b(Red)/b(Blue) | 0.47 | 0.43 | b(Blue), b(Red) | 0.41 | 0.55 |
| b(Red)/b(Green) | 0.43 | 0.42 | b(Blue), b(Green) | 0.39 | 0.48 |
| b(*NIR*)/b(Blue) | 0.47 | 0.33 | b(Coastal), b(Green) | 0.39 | 0.42 |
| b(*NIR*)/b(Green) | 0.42 | 0.30 | b(Red)/b(Blue) | 0.38 | 0.45 |
| b(SWIR)/b(Blue) | 0.14 | 0.04 | b(Green), b(Red) | 0.35 | 0.51 |

The BPNN model was trained on the Landsat-5 TM dataset. The visible- and *NIR*-band data were selected as the inputs. The output in this network is $C_{DIN}$ and $C_{PO4\_P}$. The modeling work was conducted as follows. (1) Determine the structure of the network. The simplest three-layer network structure was used. The number for the hidden layer was set to 4–10. (2) Choose an activation function. The Levenberg–Marquardt method was selected as the activation function. (3) Validation. Training was performed 5000 times to yield a result with the smallest RMSE [30]. The same training procedure was used for $C_{PO4\_P}$. The performances of the NN models were determined based on the results obtained from the independent validation set (see Figure 7). The $C_{DIN}$ and $C_{PO4\_P}$ retrieval models yielded $R^2$ values of 0.90 and 0.82 and RMSEs of 0.458 and 0.075, respectively.

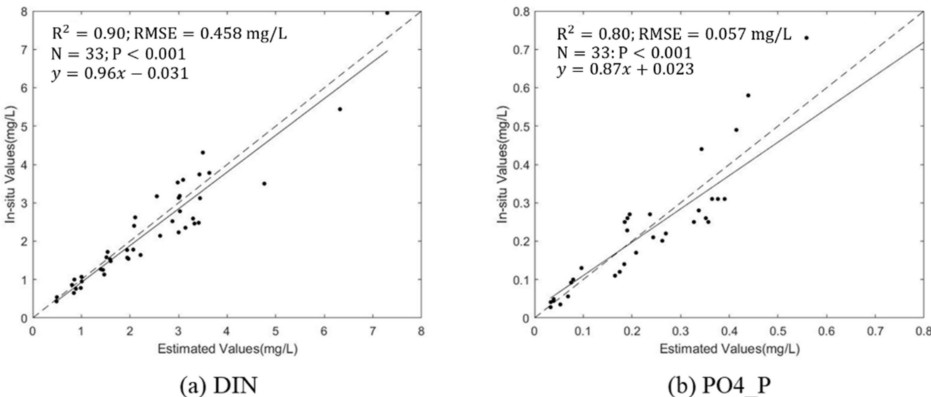

(a) DIN        (b) PO4_P

**Figure 7.** Scatterplots of the validation data for Landsat-5. (**a**) $C_{DIN}$. (**b**) $C_{PO4\_P}$.

The SVM model was trained on the Landsat-8 OLI data. The visible- and *NIR*-band Landsat-8 data were selected as the input. Similar to the neural network, the data were divided into a training set (80%) and test set (20%). To avoid overfitting, the data were subjected to a five-fold cross-validation [32]. The obtained $C_{DIN}$ and $C_{PO4\_P}$ retrieval models yielded RMSEs of 0.571 and 0.032 and $R^2$ values of 0.66 and 0.80, respectively. Due to the relatively small amount of matched Landsat-8 OLI data, Landsat-8 OLI data for the measurement months were used to form an independent validation set. The validation results are shown in Figure 8.

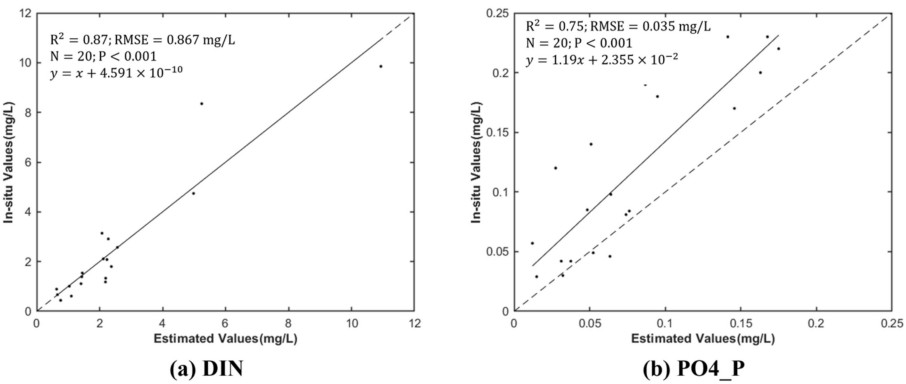

**(a) DIN**                                                                                                      **(b) PO4_P**

**Figure 8.** Results obtained from the validation data for Landsat-8. (**a**) $C_{DIN}$. (**b**) $C_{PO4\_P}$.

## 3.2. Temporal and Spatial Distributions of Concentration Retrievals

In the study area, cloud cover tends to be extensive in the summer, inhibiting construction of an RS-imagery-based time series. Thus, relatively continuous imagery in the fall and winter was selected for a long-term time-series analysis. Imagery for some years was low quality and omitted. The retrieved surface $C_{DIN}$ and $C_{PO4\_P}$ in SZB are shown in Figures 9 and 10, respectively.

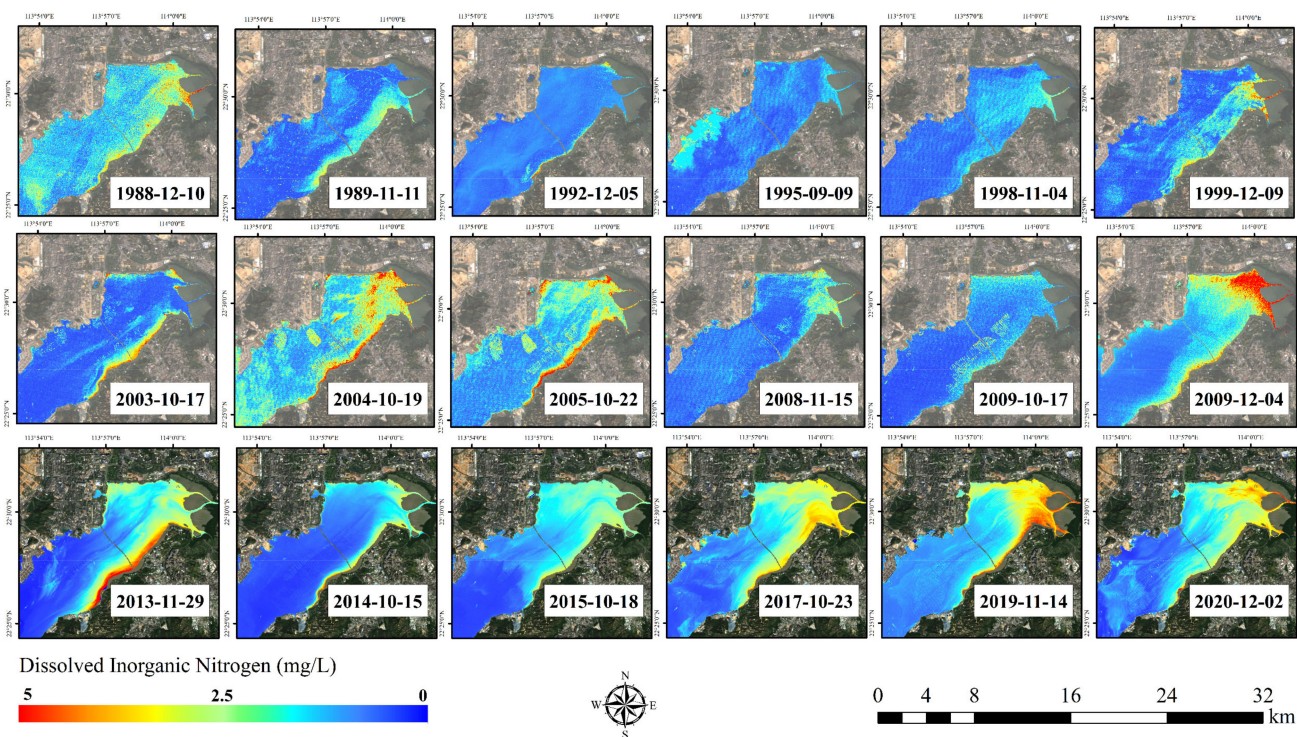

**Figure 9.** Estimated $C_{DIN}$ in SZB from RS imagery.

In Figures 9 and 10, the top, middle, and bottom rows of the retrieved images correspond to the pre-2000 period, the period between 2000 and 2010, and the post-2010 period, respectively. The retrieved multi-year nutrient concentrations show specific spatial distribution trends. The nutrient concentrations were higher in Inner SZB than Outer SZB. High concentrations were clustered in the estuary of the Shenzhen River near northeastern SZB, as well as in the areas of SZB near land. The concentrations were lower in central SZB than in the SZB regions near land, mainly because pollutants in SZB originated primarily from the rivers (e.g., the Shenzhen River and Shan Pui River) flowing into Inner SZB [19]. Pollutants produced by human activities flow into the river through surface runoff and underground pipelines. The smaller tributaries flow into the Shenzhen River and the Shan

Pui River, and finally into the SZB; this pattern greatly influences the water quality of Inner SZB. To some extent, SZB can be considered an independent system (simplified as a southwest–northeast gulf system) (Figure 11b). The pollution sources are mainly located in the northeast corner and spread to the southwest.

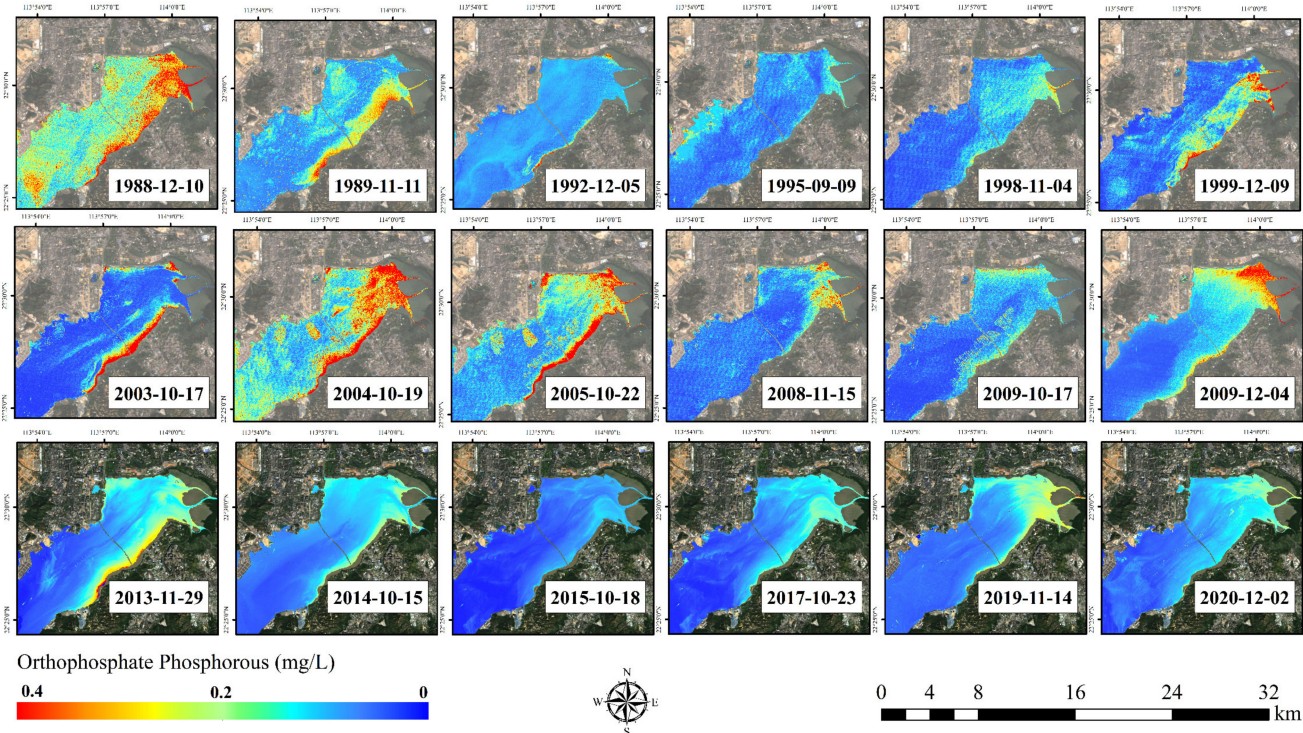

**Figure 10.** Estimated $C_{PO4\_P}$ in SZB from RS imagery.

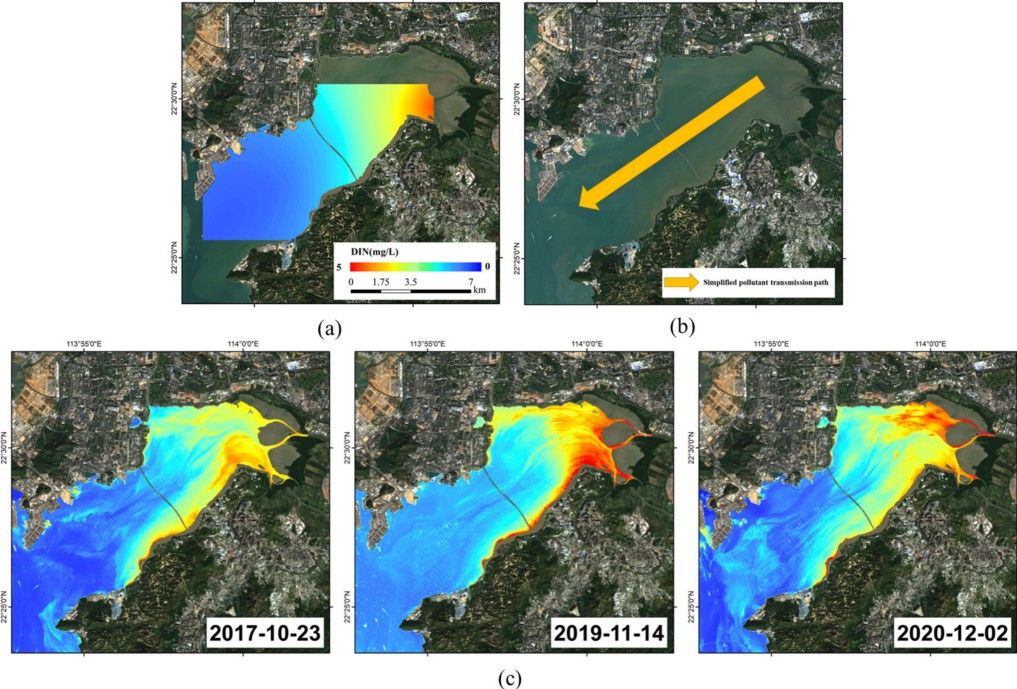

**Figure 11.** Spatial concentration of $C_{DIN}$. (**a**) Interpolation results based on multi-year average of in situ $C_{DIN}$ (from 1986 to 2019). (**b**) Simplified model of pollutant diffusion in Shenzhen Bay. (**c**) Estimated $C_{DIN}$ in SZB.

To compare the characteristics of the spatial distribution of nutrients in SZB, representative areas from Inner and Outer SZB were analyzed (see Figure 2 for locations). A comparison of the nutrient concentrations in Inner SZB and Outer SZB shows the following. The multi-year average $C_{DIN}$ was 4.17 mg/L in Inner SZB, nearly six times that (0.698 mg/L) in Outer SZB. The multi-year average $C_{PO4\_P}$ in Inner SZB was 0.297273 mg/L, nearly nine times that (0.034 mg/L) in Outer SZB. According to the Chinese Sea Water Quality Standards (GB 3097-1997) [3], the Class IV $C_{DIN}$ and $C_{PO4\_P}$ standards are 0.5 and 0.045 mg/L, respectively. A comparison shows that $C_{DIN}$ was 8.34 times higher than the standard in Inner SZB and was at the inferior Class IV level in Outer SZB; also, $C_{PO4\_P}$ in Inner SZB was 6.6 times higher than the standard [3]. These findings are consistent with the assessment of seawater quality in the annual reports published by the HKEPD, namely, the water quality in SZB was relatively poor [25].

### 3.3. Time-Series Analysis

Considering that the effectiveness of remote-sensing images in fall and winter is higher than that in spring and summer, the RS imagery for fall and winter was selected for the time-series statistics. The inversion results for Inner SZB and Outer SZB in various years are shown in Figure 12. The concentrations in Outer SZB are relatively low and mostly stable over time, with only a slight upward trend from 1988 to 2020. However, the nutrient concentration in Inner SZB greatly fluctuates, showing a gradual increase (1988–2009) and then a gradual decrease, with occasional peaks.

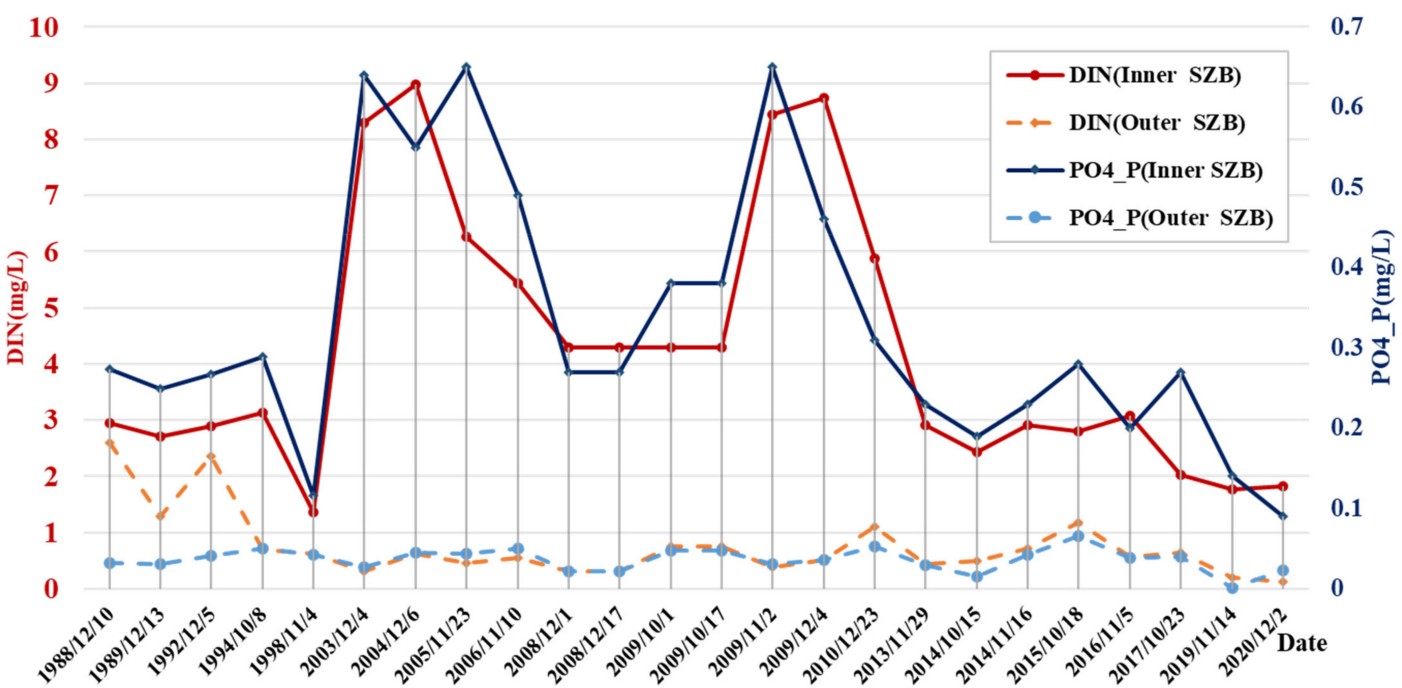

**Figure 12.** Graph of changes in $C_{DIN}$ and $C_{PO4\_P}$ in Inner SZB and Outer SZB during fall and winter.

Regression analysis of the changes of $C_{DIN}$ and $C_{PO4\_P}$ in Inner SZB and Outer SZB is shown in Table 3. A slope of less than 0 indicates a decrease and vice versa. A greater absolute value of the slope indicates a more significant change. On this basis, it can be seen that changes were more pronounced in Inner SZB than in Outer SZB. In addition, $C_{DIN}$ changed more significantly than $C_{PO4\_P}$ in Inner SZB. In contrast, in Outer SZB, the changes of $C_{DIN}$ and $C_{PO4\_P}$ were relatively insignificant, with the concentrations remaining nearly the same and $C_{DIN}$ slightly increasing.

**Table 3.** Regression analysis results for changes in $C_{DIN}$ and $C_{PO4\_P}$.

| Parameter | Location | Slope | Intercept | $R^2$ |
|---|---|---|---|---|
| DIN | Inner Bay | −0.0766 | 5.2072 | 0.0531 |
| | Outer Bay | −0.0422 | 1.2784 | 0.2443 |
| PO4_P | Inner Bay | 0.0077 | 0.4246 | 0.1101 |
| | Outer Bay | −0.0004 | 0.0412 | 0.0351 |

## 4. Discussion

### 4.1. Monitoring Water Quality of the Rivers Flowing into SZB

In total, six rivers flow directly into SZB: five from Shenzhen (Shenzhen, Fengtang, Xiaosha, Dasha, and Houhai Central River) and one from HK (Shan Pui River). Previous studies have shown that these rivers transport terrestrial pollutants and directly cause water-quality deterioration in SZB [19,20]. With estuaries of approximately 200 m (Shenzhen River) and 110 m (Shan Pui River) in width, these rivers situated in the innermost bay area are the primary sources of terrestrial pollution. A time-series graph was produced based on the data from the estuaries, as shown in Figure 13. In SZB, there are an average of two images per year (Table 1), but the climates differ from year to year resulting in a non-uniform distribution.

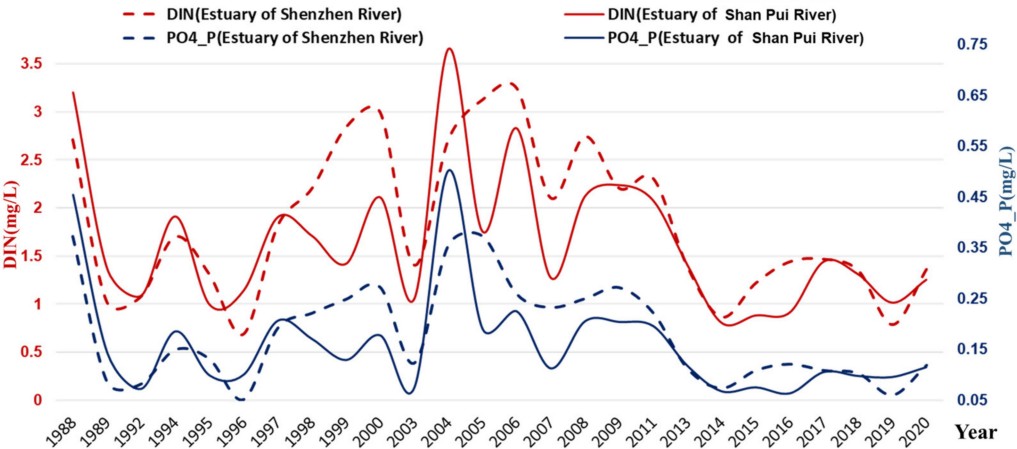

**Figure 13.** Time series of the estimated $C_{DIN}$ and $C_{PO4\_P}$ of the Shenzhen River and Shan Pui River estuaries. The time-series polyline is smoothed.

The annual average $C_{DIN}$ and $C_{PO4\_P}$ of the Shenzhen River and Shan Pui River estuaries in Figure 13 show that the nutrient concentrations first increased and then decreased; the overall trend was downward. Note the pivot points occurred at different times. The nutrient concentrations in the Shan Pui River decreased between approximately 2005 and 2009. During this period, Hong Kong focused on pollution sources through a voluntary surrender scheme for poultry and pig farming licenses. As a result, the water quality in the Shan Pui River and other surface sources that discharge into the Shenzhen River improved [5,23]. A notable decrease in the nutrient concentrations on the Shenzhen side of the SZB began after 2013. In 2000, the two governments formulated the "Deep Bay SZB Water Pollution Control Joint Implementation Program" (JIP) to improve SZB's water quality. The JIP outlines pollution-control measures to be undertaken by both governments at various stages to reduce wastewater discharge into Deep Bay by extension and improvement of sewerage infrastructure [5]. Multi-year observations indicate that the work conducted in the Shenzhen River was highly effective and that both $C_{DIN}$ and $C_{PO4\_P}$ in the Shenzhen River were declining. Prior to 2013, the diversion of rain and sewage water was relatively unsatisfactory in Shenzhen. As a result, during heavy rainfall events, rain and sewage water would overflow and directly discharge into the rivers that eventually emptied into SZB,

resulting in deteriorating water conditions in SZB. With completion of the 12th "Five-Year Plan" and the implementation of the 13th "Five-Year Plan", a sewage system based on the diversion of rain and sewage water, as well as pollution interception and treatment, has been put into operation to strengthen the comprehensive management of drainage [33]. As a consequence, there has been a decrease in overflow and direct discharge as well as a gradual improvement in the water conditions in the rivers flowing into SZB.

To further verify the inversion results, the annual average change of the monitoring point DM1 closest to the two estuaries was chosen as the reference standard, as shown in Figure 14.

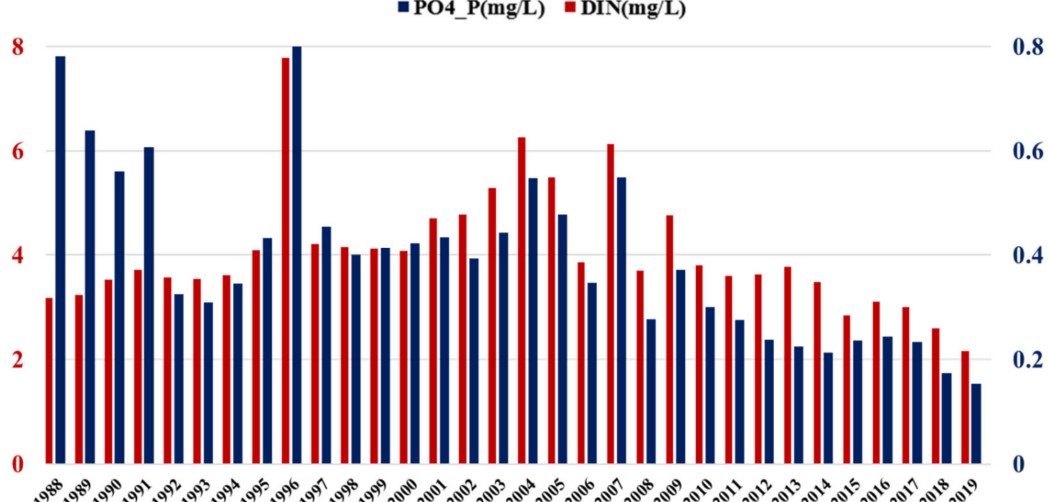

**Figure 14.** The annual average in situ values at the DM1 monitoring point from 1988 to 2019.

From 1988 to 2020, in situ $C_{DIN}$ measurements at the DM1 monitoring site showed an upward trend from 1988 to 2003, and then a downward trend after 2007; the only exception occurred in 1996. In situ $C_{PO4\_P}$ was slightly different from $C_{DIN}$. The main difference was that during 1988–1991, $C_{PO4\_P}$ was higher and then began to decrease. After 1992, $C_{PO4\_P}$ was consistent with $C_{DIN}$ and began to gradually increase, peaking in 2007, then gradually decreasing. The nutrient-concentration trend in the estuary obtained by the inversion in Figure 13 is nearly the same as that of the actual monitoring data. The overall trend is upward, then downward. The peak appeared around 2004, with a more obvious decline after 2009. Although the overall trends are similar, some differences exist, probably due to (1) the small number of images per year and the bias towards fall and winter, and (2) slight differences between the measurements at the estuaries and measurements at DM1, which is slightly farther away from the estuaries.

### 4.2. Impact of Oyster Rafts

In this study, many oyster rafts (ORs) were found close to the Hong Kong side of SZB, particularly in the Lau Fau Shan region at the interface between fresh and salty water (an area highly suitable for oyster farming). According to the information published by the Agriculture, Fisheries and Conservation Department of Hong Kong, oyster farming has been continually conducted in the intertidal mudflats along the coast of Deep Bay (SZB) for more than two centuries. However, the breeding method of using ORs had only been introduced in the past 10 years. In previous studies, researchers found that oyster farming had a low impact on local water quality [34–36]. We verified this claim as follows. In this study, the OR-distribution area was extracted to determine any changes and to investigate whether these changes were possibly correlated with the nutrient concentrations. Because ORs had only been in use for approximately 10 years, our research focused on Landsat-8 series images with a lower signal-to-noise ratio and higher quality.

The ORs significantly differ from water bodies in remote-sensing imagery but are spectrally similar to land. Therefore, ORs and water bodies can be differentiated using the *NDWI* [28]. The ORs identified for each year are shown in Figure 15.

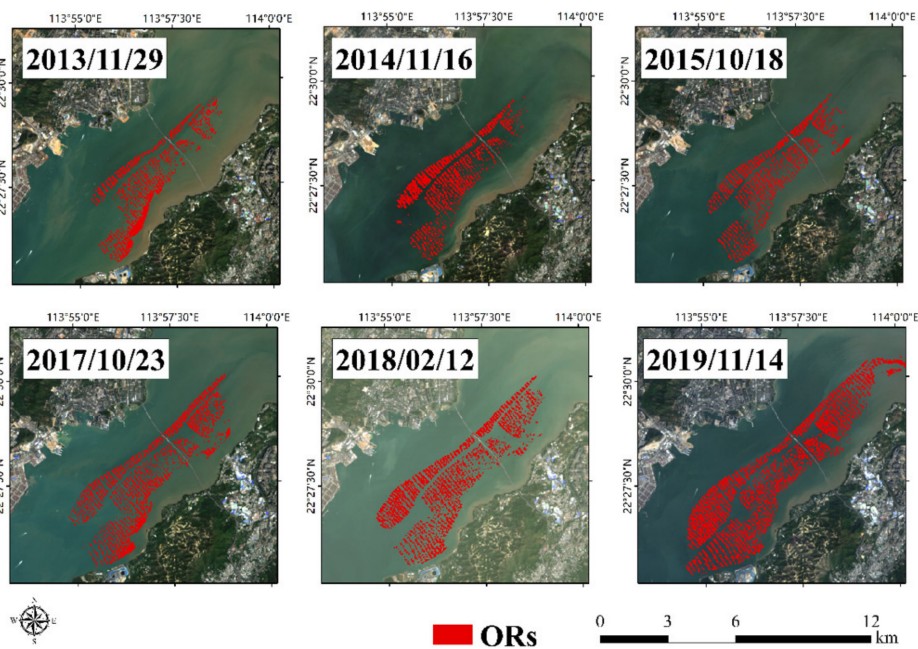

**Figure 15.** The distribution of ORs in SZB for each year.

From 2013 to 2019, the general trend is an increase in the area used for ORs. To examine the impact of ORs on water quality, a graph of annual changes in nutrient concentrations at point A (see Figure 16 for location) within the OFA in central SZB was produced (see Figure 17 for results).

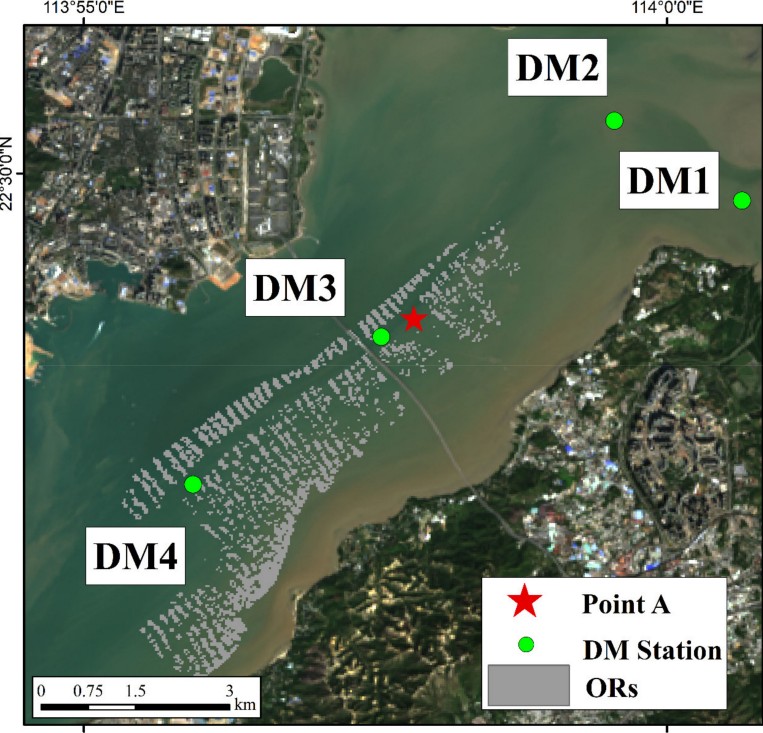

**Figure 16.** Analysis location for research on impact of oyster farming (Point A).

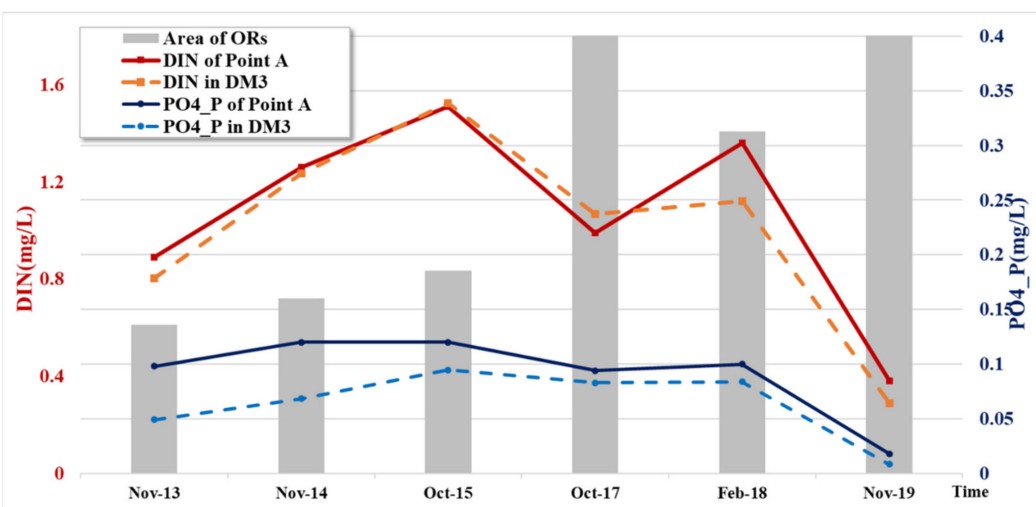

**Figure 17.** Changes in nutrient concentrations in the OFA in SZB (in situ measurements at the DM3 station versus retrieved values). The gray histogram shows the relative changes in the OFA statistical area.

During 2013–2019, there was an increase in the number of ORs, whereas the nutrient concentrations decreased, resulting in a year-to-year improvement in the water quality in the OFA in SZB. Figure 17 shows the changes in the measured nutrient concentrations at the HKEPD DM3 measurement point in the same month as the RS image, which also shows a decrease. However, whether oyster farming improved the water quality in SZB cannot be determined based solely on the results in Figure 17. Therefore, two transects along the oyster rafts were selected for further research: one transect was near the ORs (Transect A in Figure 18) and the other was 1000 m from the ORs (see Transect B in Figure 18). The multi-year-average nutrient concentrations from 2013 to 2019 in the waters near OFA and approximately 1000 m from OFA were further compared (see Figure 19 for results).

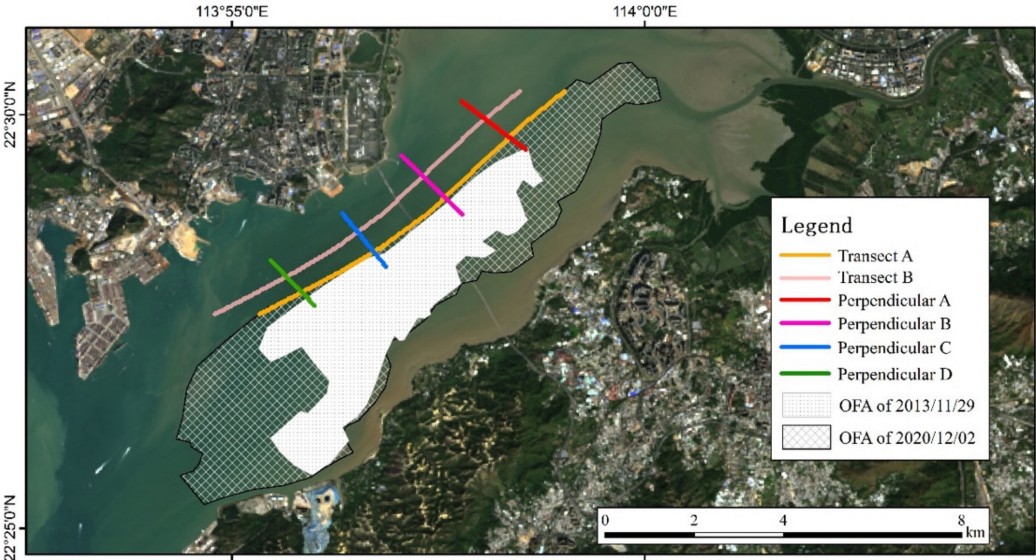

**Figure 18.** Transect and perpendicular lines for researching the impact of oyster farming on water quality. Transect A is near the ORs, while Transect B is approximately 1000 m from the ORs.

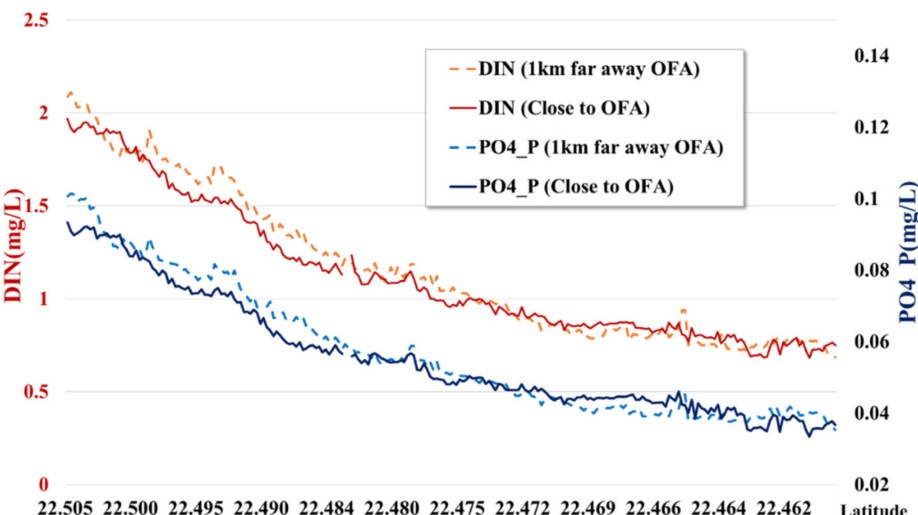

**Figure 19.** The multi-year-average nutrient concentrations near to and 1000 m from the OFA from 2013 to 2019.

In the graph in Figure 19, the latitude gradually decreases from left to right, corresponding to the direction from Inner SZB to Outer SZB. Based on Figure 10, the nutrient concentrations were significantly reduced from Inner to Outer SZB. The difference between the nutrient concentrations near OFA and 1000 m away is not obvious, but the value does fluctuate. However, as the distance from OFA decreases, the water quality neither improves nor deteriorates. Based on these results, it is not clear whether oyster farming improves the water quality of SZB because oyster farming is not the only influencing factor.

The nutrient concentrations at the four perpendicular lines (see Figure 18 for locations) are shown in Figure 20. To avoid the influence of land-based river input on the south side of the ORs, four perpendiculars were selected on the north side of the ORs and away from the land. The solid line represents the overlapping area with the ORs. As the latitude increases, the perpendicular overlaps with the OFA, and the nutrient concentrations tend to decrease slightly.

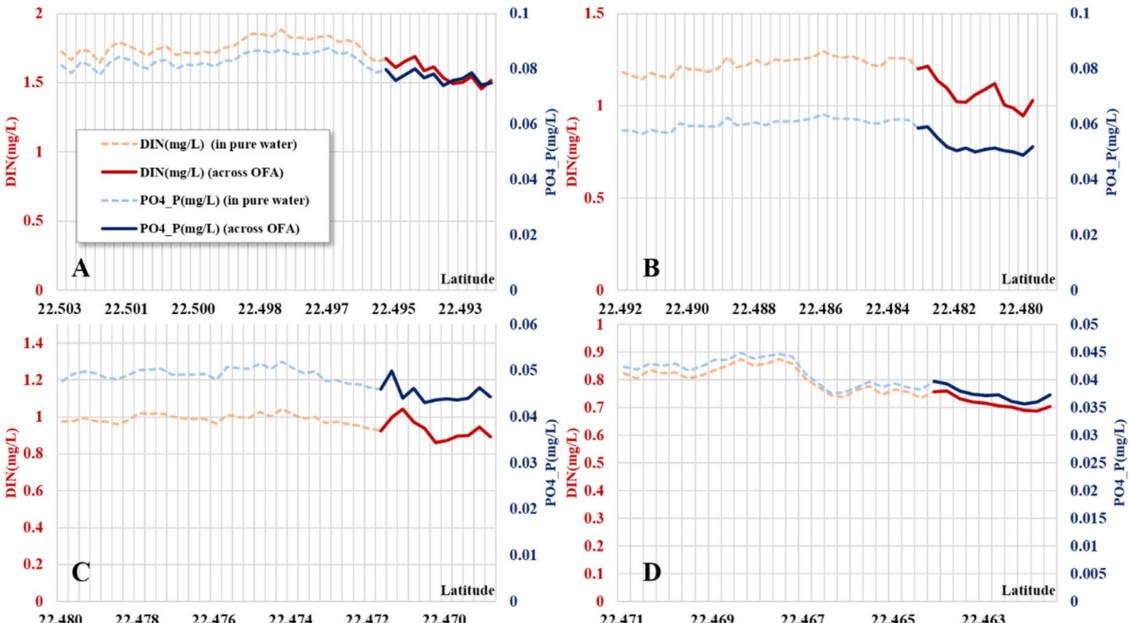

**Figure 20.** Nutrient concentrations at four perpendicular lines. (**A–D**) represent the results of the same four perpendiculars in Figure 18.

The appearance of ORs did not significantly change the basic spatial distribution of nutrient concentrations in SZB. A higher nutrient concentration was not found in the OFA, and the high nutrient concentration was still concentrated near the land and at the mouth of the river.

## 5. Conclusions

Offshore aquatic environments are complex. As a result, simple linear relationships cannot be used to analyze spectral information and WQFs. In comparison, nonlinear models are very broad in scope and can be continuously optimized through multiple training times based on ML. This aspect allows independent variables to increasingly approximate true dependent variables through nonlinear relationships. Different ML methods were trained based on the characteristics of the Landsat-5 TM and Landsat-8 OLI data that matched the in-situ data. Using the BPNN method, the validation shows that the final Landsat-5 TM-based $C_{DIN}$ and $C_{PO4\_P}$ retrieval models yielded $R^2$ values of 0.90 and 0.82 and RMSEs of 0.458 and 0.075, respectively. A second-order SVM ($R^2 = 0.66$, RMSE = 0.571) was selected to retrieve $C_{DIN}$ from Landsat-8 OLI data. A second-order SVM model with a visible band combination as the input ($R^2 = 0.80$, RMSE = 0.032) was selected to retrieve $C_{PO4\_P}$ from Landsat-8 OLI data.

In terms of the overall spatial distribution, the water quality in Inner SZB was worse than that in Outer SZB. Specifically, $C_{DIN}$ and $C_{PO4\_P}$ values in Inner SZB were 2–9 and approximately 4–19 times higher, respectively, than those in the Outer SZB. From Inner to Outer SZB, the decrease in nutrient concentration is driven by the inflow of the Shenzhen River. This conclusion is consistent with the land-based metal enrichment proposed by Liu et al. due to anthropogenic pollutants discharged by riverbank runoff [37]. Outer SZB is farther away from the Shenzhen River, and it is connected to the open sea such that the water exchange capacity is stronger than that of Inner SZB. With these factors, the concentrations of nutrients in Outer SZB are lower than those in Inner SZB. In the central SZB, there were a large number of ORs, which had a relatively insignificant impact on the overall water quality. During 1988–2020, the concentrations of each WQF in Inner SZB fluctuated and eventually displayed an upward and then a downward trend. In Outer SZB, the nutrient concentrations remained relatively stable and exhibited downward trends with relatively small slopes. The water quality in SZB was primarily affected by the rivers. Thus, the water conditions in the estuaries of several key rivers were determined based on remote-sensing data. The results show the following. A series of restoration operations, particularly those targeting the Shenzhen River, improved the quality of the water discharged into SZB [25]. $C_{DIN}$ decreased to approximately half of the pre-2013 multi-year average, while $C_{PO4\_P}$ also decreased from the pre-2013 levels. These improvements were closely related to the measures implemented in Shenzhen after 2013, such as comprehensive strengthening of drainage management and diversion of rain and sewage water [33].

This study demonstrates the feasibility of remote sensing in monitoring water quality in bays. The use of remote-sensing data to obtain large-area coverage and long-term traceable water quality data is conducive to the supervision and monitoring of the bay. At present, many countries and regions have launched ecological governance work for gulfs. The application of remote-sensing technology to gulf water quality can support the traceability of pollution sources; it can also evaluate, supervise, and track the process of gulf governance projects. Overall, remote-sensing technology provides a new way of studying a bay's ecological environment, and it helps promote ecological management.

However, the current research still has issues, such as the small amount of useful data and underestimating instantaneous tidal effects. Future research will attempt to combine multiple satellites with higher spatial and temporal resolutions, for example, matching the higher spatial resolution satellite Sentinel-2 with measurement data to compare the pros and cons of different inversion results. The use of inversion algorithms still needs to be explored in terms of modeling methods for small sample data and algorithm optimization.

**Author Contributions:** Conceptualization, X.H. and D.W.; methodology, J.H.; software, validation, J.H.; formal analysis, J.H. and F.G.; resources, F.G. and D.W.; writing—original draft preparation, J.H.; writing—review and editing, D.W., X.H. and Y.B.; supervision, project administration and funding acquisition, D.W. All authors have read and agreed to the published version of the manuscript.

**Funding:** This work was supported by the National Key R&D Program of China under Grant No. 2018YFB0505005 and 2017YFC1405300, the Key Research and Development Plan of Zhejiang Province under contract No. 2017C03037, and the National Natural Science Foundation of China under contract No. 41476157.

**Acknowledgments:** The authors would like to thank the USGS for excellent Landsat data and HKEPD for the valuable measurement data. We also thank the satellite ground station and the satellite data processing & sharing center of SOED/SIO for help with the data processing. Our deepest gratitude goes to the editors and reviewers for their careful work and thoughtful suggestions.

**Conflicts of Interest:** The authors declare no conflict of interest.

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
