# Peer review of "Changes in Nutrient Concentrations in Shenzhen Bay Detected Using Landsat Imagery between 1988 and 2020"

_remotesensing, doi:10.3390/rs13173469_

Round 1

Reviewer 1 Report

SUMMARY

The paper addresses the research area related to the monitoring of key water-quality factors in Shenzen Bay in the period 1988-2020 through remote sensing data.

 It aims to assess the concentrations of dissolved inorganic nitrogen (DIN) and orthophosphate-phosphorus (PO4_P) by means of long Landsat time-series data and machine-learning algorithms.

The authors claim that the water quality in the river estuaries in SZB was affected mainly by river input.

Abstract

Please, consider improving the quality and structure of the abstract.

MINOR COMMENTs

Minor corrections and text editing.

L20 Please, consider introducing before the “oyster-farming” item.

L107. Please, clarify “Case-2”

L161-166. Please, consider inserting a table/list of images (acquisition time, path, row, platform, etc) considered in this study. It would be useful to guarantee the replicability of these experiments.

L169 Please, consider inserting more details and references for the “dual-band threshold method”.

L175. Please, consider improving or at least removing the 2.4 section (Evaluation indexes)

L187 “To ensure synchronicity, the remote-sensing images acquired within five days of the measurement dates were selected”

Please consider repeating the experiments by using RS images strictly synchronized with ground measurements.

In the alternative, please consider inserting references that can confirm that the variation of CDIN e CPO4_P is negligible across more days.

Author Response

Thank you very much for your valuable comments. Please see the attachment.

Reviewer 2 Report

Derivation of water chemistry data via remote sensing of optically inactive components is challenging but is worth attempting given the advantages in terms of cost and area coverage which satellite data can provide. This manuscript presents one such attempt which does so via use of machine learning approaches. In general, the manuscript is well written and the argument is straightforward to follow. However, there are a number of inconsistencies, unclear sections and areas of weakness which need to be remedied before it is suitable for publication in an international peer-reviewed journal.  Consequently, I am recommending major revisions. I provide below a summary of my main comments. More detailed recommendations and specific corrections can be found in the attached pdf.

1) Introduction

Generally well written, only minor edits and slight restructuring required.

2) Materials and methods

Generally OK, some areas where clarification is needed. Also, some material from section 3.1 should be moved into this section. I admit I am not an expert on ML approaches, therefore I cannot comment on the technicalities of the algorithms used and whether they are appropriate.

3) Results

Section 3.1: multiple areas which are unclear or need additional information.

Section 3.2: there are some additional spatial patterns and reasons for these which need to be mentioned in this section. Some small passages of text are unclear.

Section 3.3: has inconsistencies with respect to the presented time series data, particularly Figure 11 vs Figures 8-9. There is either a mistake here, or unclear explanation of method. Additional written description of the time-series patterns shown is needed.

4) Discussion

Restructuring is suggested, effectively swapping sections 4.1 and 4.2 so that discussion considers first the general pattern in terms of nutrient sources (estuaries) and nutrient concentration change over time, and then finishes with the more focussed oyster bed case study. Both sections however need more work.

Section 4.1: This section feels weak and something like an exploratory add-on. I have provided some suggestions for stronger theoretical justification. There is also a major omission in terms of using a single unjustified timestep as a proxy for spatial patterning (Figure 15) which must be explained.

Section 4.2: This section is weakened by its reliance on averaged and temporally very discontinuous satellite images. Therefore, I advise inclusion of a time-series of in situ water quality measurements for comparison and validation. Effectively, there needs to be an independent comparison of water quality change over time against which the satellite-derived data can be compared. More clarity is also needed as to how much averaging was done (e.g. how many Landsat images were averaged for each year?). I make these suggestions because whilst satellite-derived data are great in terms of spatial coverage, their temporal resolution can be relatively poor due to issues such as cloud cover, and this can be a major disadvantage when trying to use them to obtain data on rapidly changing and complex environments, such as coastal waters. More citations are also needed to provide evidence for the statements on pollution action/changes in the study area which are described.

Figures

Generally clear, but please check the Figure numbers. Several are incorrect. Several figures also have typos and/or could be improved.

Author Response

(The authors gave the same response as above.)

Reviewer 3 Report

Generally, I have not critical remarks and in my opinion, the paper is suitable for inclusion in the scientific journal Remote Sensing after minor correction. The overall manuscript is well written and it is nice to be read. The structure of the manuscript follows the scientific outline, however, I suggest to add a Motivation (or Contribution) section. Please, explain in the Motivation section, why have you conducted such research. Some sentences from the Introduction may have to be moved to the Motivation section.

Additionally: some of the References lack DOI and number of pages, please add if available.

Author Response

(The authors gave the same response as above.)

Reviewer 4 Report

Overall, the manuscript is interesting and the results are promising. However, the authors should describe better the rationale of the study and the aim of the paper.

I suggest a slight change of the title to be more attractive:

Changes in nutrient concentrations in Shenzhen Bay detected using remote-sensing data between 1988 and 2020

A better description of the forecasting algorithms (BPNN and SVM) should be considered. See an example regarding the description of the steps in DOI: 10.5772/63109

The time series from figure 5 seem to be smoothed. Please describe this aspect in the text.

Please include in the discussion the limitations and future work.

Describe better the impact of your findings on the future management of the bay to maintain sustainability and water quality.

English language should be further polished to increase readability. 

Author Response

(The authors gave the same response as above.)

Round 2

Reviewer 1 Report

The paper has been improved since the first review round.

Minor corrections and text editing are needed.

Author Response

Dear reviewer,

Thank you very much for your previous suggestions. According to the opinions of other reviewers, we have made more modifications. I hope you are also satisfied with the revised manuscript.

In addition, the article has been polished in English. Since there are many traces of modification after English polishing, the red text is used to mark the places modified according to reviewer opinions.

Thanks again.

Reviewer 2 Report

Thank you for taking my comments on board. With the additions, clarifications and re-structuring, the paper now reads better and is easier to follow. However, there are still some changes which need to be made. As before, my general comments are here with more detailed examples and specific corrections on the attached pdf:

1) Unfortunately, the revisions have resulted, in places, in typos and poor English. I have identified several examples of these and have suggested corrections. Note that these suggestions are not exhaustive. Therefore, I also suggest that whichever of the authors has the best English language skills takes a final read though the entire article before it is resubmitted, to ensure that all the language is correct.

2) Section 4.1 – thank you for including the direct comparison with in situ data, it makes for good comparison and validation. However, the comparison is not so simple as the current text implies, and there are some differences which need to be explained. I have left a detailed comment to this effect on the pdf.

3) Section 4.2 – the nature of the comparison is still not clear – are you comparing areas or transects? The maps/images imply area, but the graph implies a transect. It is still also unclear what timestep is used due to inconsistencies in the text. Again, see detailed comment on the pdf.

Once these issues are sorted out, I would be happy to recommend this for publication.

Author Response

Dear reviewer,

Thank you very much for your suggestions. Please see the attachment.

In addition, the article has been polished in English. Since there are many traces of modification after English polishing, the red text is used to mark the places modified according to reviewer opinions.

Thanks again.

Reviewer 4 Report

Following the first round of revision. the authors have addressed my technical observations. There are some paragraphs that still require further English polishing and if possible one last editing should be performed before resubmission. During this process check also the punctuation.

Author Response

(The authors gave the same response as above.)
